# Unsupervised Anomaly Detection for Auditing Data and Impact of Categorical Encodings

**Ajay Chawda**
Department of Computer Science
TU Kaiserslautern
Financial Mathematics, Fraunhofer ITWM
`a_chawda19@cs.uni-kl.de`

**Stefanie Grimm**
Financial Mathematics
Fraunhofer ITWM
Kaiserslautern DE
`stefanie.grimm@itwm.fraunhofer.de`

**Marius Kloft**
Department of Computer Science
TU Kaiserslautern
Kaiserslautern DE
`kloft@cs.uni-kl.de`

## Abstract

In this paper, we introduce the *Vehicle Claims* dataset, consisting of fraudulent insurance claims for automotive repairs. The data belongs to the more broad category of **Auditing** data, which includes also *Journals* and *Network Intrusion* data. Insurance claim data are distinctively different from other auditing data (such as network intrusion data) in their high number of categorical attributes. We tackle the common problem of missing benchmark datasets for anomaly detection: datasets are mostly confidential, and the public tabular datasets do not contain relevant and sufficient categorical attributes. Therefore, a large-sized dataset is created for this purpose and referred to as *Vehicle Claims* (VC) dataset. The dataset is evaluated on shallow and deep learning methods. Due to the introduction of categorical attributes, we encounter the challenge of encoding them for the large dataset. As *One Hot* encoding of high cardinal dataset invokes the "curse of dimensionality", we experiment with GEL encoding and embedding layer for representing categorical attributes. Our work compares competitive learning, reconstruction-error, density estimation and contrastive learning approaches for *Label*, *One Hot*, *GEL* encoding and embedding layer to handle categorical values.

## 1 Introduction

In the context of auditing data, most of the anomaly detection methods are trained on private datasets. These datasets contain personal information regarding individuals and companies, which can be used for malicious purposes by hackers in the event the knowledge becomes public. To hide the sensitive information, we need a dataset that models the behaviour of private auditing data but does not contain personal information about individuals. The lack of a benchmark dataset in the context of auditing data motivates us to create an anomaly benchmark. (Ruff et al., 2021) mentions three strategies for anomaly benchmarks: *k-classes-out*, where we consider a few classes in multiclass data to be normal and the rest as anomalous; *Synthetic*, where we use a supervised dataset and insert synthetic anomalies; and *Real World*, where we label the dataset with the help by a human expert. In this paper, we follow the *Synthetic* approach and create a synthetic dataset based on domain knowledge learned from the fraudulent claims dataset of an automotive dealer.

NeurIPS 2022 Workshop on Synthetic Data for Empowering ML Research.

Fraud in finance, insurance, and healthcare is ubiquitous. An insurance claim for a vehicle can be marked as an anomaly if it amounts to the sum of repairing an engine issue when in reality, the claim is to fix a punctured tyre, or a transaction on a credit card of the same amount as the total yearly bill for the purchase of a book will be an anomalous sample. These seemingly anomalous data points might be fraudulent. However, they may be noisy normal observations naturally varying from the default behavior and not fraudulent, which may become evident when studying them using domain knowledge. An insurance claim of an unusually high amount might be genuine, and due to the incorrect reason listed in the dataset, it results in a fraudulent claim. If there is prior knowledge about the data being an anomaly, it will make our task easier. However, in a real-world scenario, data does not come with labels, and labeling a dataset is an expensive task. Due to the unavailability of labels, we focus our attention on unsupervised methods for anomaly detection.

In anomaly detection, our goal is to distinguish between normal and anomalous samples. We use classic machine learning and modern deep learning methods for anomaly detection (AD) as baselines in this paper. Classic unsupervised methods include Isolation forest (Liu et al., 2008), which is an ensemble of binary decision trees and create shorter paths for anomalies, One-class support vector machines (OC-SVM) (Schölkopf et al., 2001), which encompasses the normal samples with the smallest possible hypersphere, and Local Outlier Factor (Breunig et al., 2000), which predicts outliers based on the local neighborhood of the sample. These methods are used as shallow unsupervised baselines. Other supervised baselines used in this paper are Random Forest (Breiman, 2001) and Gradient Boosting (Friedman, 2001). Our work evaluates RSRAE (Lai et al., 2020) for reconstruction error method, DAGMM, SOM-DAGMM (Zong et al., 2018; Chen et al., 2021a) for density estimation method, SOM (Kohonen, 1990) for competitive learning and NeutralAD (Qiu et al., 2021) and LOE (Qiu et al., 2022) for contrastive learning approaches in unsupervised anomaly detection.

Comparing deep unsupervised methods based on distinguishing concepts helps us analyze the suitable methods for auditing datasets. Further, the encoding of categorical attributes is another challenge in unsupervised learning. In the context of auditing data, there is literature associated with the evaluation of models (Nonnenmacher and Gómez, 2021; Gomes et al., 2021), but there is a lack of information on the representation of categorical features in unsupervised learning scenarios. Recent use of embeddings (Dahouda and Joe, 2021; Guo and Berkhahn, 2016; Karingula et al., 2021) encourages us to use embedding for encoding categorical features. We also use GEL (Golinko and Zhu, 2019) encoding to mitigate issues due to the One Hot representation of the high cardinality dataset. The labels are available for the datasets used in the paper, which helped us to report AUC and F1 scores. In an entirely unsupervised setting, this would not be possible, and we need to look for alternatives. The choice of evaluation metric for anomaly detection is important as noted by (Alvarez et al., 2022), so we discuss the difference between AUC and evaluating under thresholds using F1 scores.

Our work solves the problem of a missing public dataset for auditing data where categorical attributes are dominant and decisive in the classification process of normal and anomalous samples. We demonstrate the performance of the synthetic dataset for various shallow and deep unsupervised methods and show that it performs similarly to the existing datasets. We also observe that encoding categorical attributes is an essential factor in the task of unsupervised anomaly detection, where the goal is to embed the attributes into meaningful representations for the model.

## 2  Dataset

Table 1: Summary of datasets. *Auditing* data is different from Credit card dataset and arrhythmia dataset in the number of categorical features.

| Dataset | Source | Rows | Num. features | Cat. features | Anomaly ratio |
|---------|--------|------|---------------|---------------|---------------|
| Car Insurance | Kaggle | 1000 | 16 | 17 | 0.25 |
| Vehicle Insurance | Oracle | 15420 | 6 | 26 | 0.06 |
| **Vehicle Claims** | Ours | 268255 | 5 | 13 | 0.21 |
| Credit Card | MLG-ULB | 284807 | 30 | 0 | 0.001 |
| KDD | UCI | 494021 | 34 | 7 | 0.2 |
| Arrhythmia | UCI | 452 | 279 | 0 | 0.26 |

The principal component of a machine learning or deep learning model is **Data**. *Experimental data* is collected in a laboratory or controlled environment where the researcher participates in the data

collection process and can alter the outcome with the changes in variables, whereas *Observational data*, is where the researcher is an observer of the data collection process. The latter type of Data is usually a byproduct of business activities and represents the facts that have happened in the past (Cunningham, 2020). *Auditing data* commonly belong to the category of observational data. Therefore, the prior knowledge of fraudulent insurance claims would exist in the case of experimental data and help in labeling the dataset. In our work, we evaluate unsupervised methods on both experimental and observational data. The datasets used in our work have labels available, which makes our task of evaluating the models easier. The observational data for insurance claims are small and medium size datasets that are downloaded from (Kaggle, 2018; Oracle, 2017). The experimental dataset is a large dataset generated from metadata of the DVM car dataset (Huang et al., 2021). Although the availability of labels helps with the analysis of our models, the goal is to reduce dependence on them for training purposes in future scenarios because of the economic expense incurred while labeling the data. The purpose of using different datasets with various sizes and features provides us with a platform to analyze the behavior of trained models on the amount of data. The arrhythmia and KDD datasets are used as benchmarks in unsupervised learning methods (Zong et al., 2018),(Qiu et al., 2022) for anomaly detection and have fewer categorical attributes than those found in auditing datasets. As observed in Table 1, auditing datasets have more categorical features, which are important to understand the behavior of data points. Another important piece of information in the above table is the anomaly ratio. We observe that it ranges from 0.001 to 0.26, i.e., the number of positive class samples (or anomalies) in the complete dataset. It is an individual choice to consider either positive or negative class as anomalous data, and in this paper, we consider positive labels as anomalies.

## 2.1 Vehicle Claims

There are multiple tabular datasets available for anomaly detection. The available benchmark datasets for anomaly detection are Credit Card fraud (MLG-ULB, 2008), KDD, Arrhythmia, and Thyroid (Blake et al., 1998) are not suitable to conclude the performance of models on auditing data. Therefore, a new synthetic dataset is created with the domain knowledge about insurance claims of vehicles for the auditing scenario. (Huang et al., 2021) released the DVM-Car dataset, which consists of 1.4 million car images from various advertising platforms and six tables consisting of metadata about the cars in the dataset. Using a Basic table, mainly for indexing other tables, which includes 1,011 generic models from 101 automakers, a Sales table, sales of 773 car models from 2001 to 2020, a Price table, basic price data of historical models, and a Trim table, includes 0.33 million trim-level information such as selling price, fuel type and engine size, the new dataset is created with 268,255 rows. This dataset is referred as Vehicle Claims (VC) dataset.

A fraudulent sample will have an insurance amount claimed that is higher than the average claim amount for the respective case. For example, a tyre is replaced by the customer due to a puncture, but the claim amount reflects the cost of replacing two tyres. This is a fraudulent claim and an anomaly. Our idea is to model this information in the dataset with the following steps. *Firstly*, categorical features *issue*, *issue_id* are added. *issue_id* is a subcategory of the issue column. *Secondly*, the *repair_complexity* column is added based on the maker of the vehicle. The common brands like Volkswagen have complexity one whereas Ferrari has four. *Thirdly*, *repair_hours* and *repair_cost* are calculated based on the *issue*, *issue_id* and *repair_complexity*. Every tenth row in *repair_cost* and $20^{th}$ row in *repair_hours* is an anomaly. *Lastly*, Labels are added for the purpose of verification. There are 56,749 anomalous points in the VC dataset. The importance of having anomalous values in categorical features like *door_num*, *Engin_size*, etc. can be argued as, for insurance claims, numerical features such as *Price* are more important, but the categorical attributes help to observe the amount of bias added to the model by low importance features, and also the categorical anomalies will be helpful for explainable anomaly detection models.

**Issues** are randomly assigned from a list of existing known issues about the vehicles. **Issue_Id** is a subcategory of certain issues. *Warning Light* has 8 sub-categories because a warning light can mean anything ranging from low fuel or engine oil to a damaged braking system. Similarly, *Engine Issue*, *Transmission Issue*, and *Electrical Issue* have 4,3, and 5 sub-categories. The list of issues contains the values as follows.

The **repair_complexity** column contains the complexity of repairing a vehicle depending upon the availability of the workshop and the price of the vehicle. Common automotive makers like

Volkswagen have complexity 1, whereas Ferrari has complexity 3. The complexity of the models and the respective makers are listed in Table 4.

The **repair_hours** attribute is the required time to repair the vehicle which is calculated by multiplying the *repair_complexity* of the vehicle with a predefined number of hours for the *issue* and *issue_id*. The **repair_cost** is the sum of **repair_hours** times 20 plus a fraction of the price of the vehicle. We have assumed the hourly work rate of labor to be 20 units. The anomalies are introduced at every $10^{th}$ instance in **repair_cost** and lie between 3 and 6 standard deviations from the mean. Every $20^{th}$ row in **repair_hours** attribute is an anomaly and lies between 3 and 4 standard deviations from the mean. This ratio can be modified to suit the problem statement. If the goal is anomaly detection in an imbalanced setting, the anomalies can be diluted to a ratio of 0.01. Table 3 lists the number of hours and cost of repair for respective **Issue** and **Issue_Id**.

**Missing** values in the table are replaced by anomalous values. Our idea was to introduce categorical anomalies in the dataset. In the real world, most of the time due to the filling of incorrect information in healthcare or insurance forms, it is difficult to find out the reason for an anomalous sample. We believe these missing values being replaced by anomalous values will be useful for explainable models to determine the cell values due to which the data point behaves as an anomaly. The missing values in each attribute of the original data are replaced as follows.

*Color*: Gelb, *Reg_year*: 3010, *Body_type*: Wood, *Door_num* : 0, *Engin_size*: 999.0L, *Gearbox* : Hybrid, *Fuel_type* : Hydrogen

There are other attributes *breakdown_date*, *repair_date* that are not used in our evaluation which contain the date of breakdown and repair date of the vehicle. The anomalous points in these attributes are the ones where there is a large difference between the two dates. If the number of hours required to repair is 9, the vehicle should be returned in 2 days, considering an 8-hour work day.

We use three insurance claims datasets to evaluate unsupervised models for auditing data. The characteristics of the VC dataset are modeled on the real-world problem of fraudulent claims. Our dataset consists of the essential features that are detrimental to distinguishing normal and anomalous samples. The number of anomalies in our dataset can be adjusted to the suitability of the training environment. The cardinality of the categorical attributes in the dataset is 1171 since it is a large dataset, and each categorical attribute contains more unique values than CI and VI datasets. We will observe that the VC dataset suffers from the curse of dimensionality

## 3  Data splitting and Evaluation metrics

In this section, we briefly describe the training and evaluation strategy of our work. We select models from different categories of unsupervised learning approaches and observe the performance of our dataset. (Ruff et al., 2021), categorizes anomaly detection methods into three topics. *First*, Density estimation and Probabilistic methods, which predict anomalies by estimating the distribution of normal data, DAGMM (Zong et al., 2018), SOM-DAGMM (Chen et al., 2021a) belong to the energy-based models under the density estimation category. *Second*, One Class Classification approaches aim to learn a decision boundary on the normal data distribution. *Reconstruction Models*, that predict anomalies based on high reconstruction error of test data. RSRAE (Lai et al., 2020) belongs to the category of deep autoencoders. Another deep learning variant is Self Organizing maps, which are trained by competitive learning instead of backpropagation learning. SOM has been recently involved in the field of intrusion detection (Chen et al., 2021a;b) and fraud detection (Mongwe and Malan, 2020). Since SOM (Kohonen, 1990) is used in SOM-DAGMM (Chen et al., 2021a) to improve the topological structure of the latent representation of the data; our aim is to investigate the results using a simple SOM. In the original paper, DAGMM and SOM are trained on network intrusion dataset (KDD) containing more numerical features the categorical features, RSRAE is the state-of-the-art model image (Fashion MNIST) and document (20NewsGroups) datasets in the field of unsupervised anomaly detection. LOE trains the model with contaminated data, which is suitable for our approach. The code and dataset are available on Github[1].

In the literature for unsupervised anomaly detection, different data splits are used to train and evaluate models. We need to decide whether to use only normal data or data with anomalous samples for training. In the latest work, (Qiu et al., 2022) provides the training framework where the model is

---

[1]https://github.com/ajaychawda58/UADAD

subjected to contaminated data. To model a real-world setting, we should train with data containing anomalies, but (Zong et al., 2018), (Zenati et al., 2018), and (Bergman and Hoshen, 2020) split the data into 50 percent normal data for training and 50 percent normal data plus anomalous data for testing. (Alvarez et al., 2022) call this as the "Recycling strategy". Another strategy is the "Discarding strategy", where first the data is split 50-50, and then the anomalies are removed from the training set as seen in (Zhai et al., 2016). The issue with this strategy is that we have less percentage of anomalous points in the test set.

As anomalous data is rare, it should not be removed from the dataset because we want to evaluate if our algorithm can find all kinds of anomalies or only of a particular kind. (Zenati et al., 2018) used the recycling strategy and compared his work with discarding strategy papers which does not provide a consistent performance comparison. Therefore, we split the train test data with a 70-30 split containing an equal percentage of anomalies. Since our work compares different encodings across models, we create separate CSV (comma-separated values) files for each encoding in the beginning. Hence, the dataset is consistent for all encodings and the comparison of algorithms is consistent. In anomaly detection, precision, recall, and F1-score are the metrics used in most literature to benchmark models. These metrics depend on the threshold on which they are calculated. We need to set a specific threshold for all models to compare different models. (Fourure et al., 2021) demonstrated that the F1-score could be manipulated by increasing or decreasing the number of anomalous samples in the test set. (Zong et al., 2018), (Zenati et al., 2018) and (Bergman and Hoshen, 2020) set the threshold $\alpha$ depending on the ratio of normal samples in the test set. In contrast, (Qiu et al., 2021) searches for an optimal threshold where the models achieve the best performance. We set the anomalies as positive classes and normal samples as negative classes. Our interest in reducing false positives and increasing true negatives motivates us to report weighted F1 scores in A.6 i.e., the metrics are calculated for both classes, and the average is weighted by the number of true samples for each class.

## 4   Results

Our idea is to use various deep unsupervised models for the evaluation but since the datasets used in this work are not reported in other literature, we use shallow supervised and unsupervised methods for baselines and compare the performance of deep unsupervised models. Appendix A.1 lists the engineered feature values of the synthetic dataset, A.2 contains details information about the encodings used for evaluation, A.3 briefly describes the models used for evaluation, and A.4 explains the use of embedding layer with autoencoders, A.5 shows our experimental setting to reproduce the results, and A.6 explains evaluation w.r.t threshold and the challenge in selecting the threshold.

Table 2: AUC scores for Car Insurance (CI), Vehicle Insurance (VI), and Vehicle Claims (VC) datasets. We observe the impact of Label (L), One Hot (O), GEL (G) encodings, and Embedding (E) layers on the baseline and deep unsupervised models. The models above the partition are deep unsupervised models, and below are the shallow baseline approaches.

| | CI | | | | VI | | | | VC | | | |
|---|---|---|---|---|---|---|---|---|---|---|---|---|
| | L | O | G | E | L | O | G | E | L | O | G | E |
| RSRAE | **55.87** | **58.12** | **58.46** | 48.26 | 55.98 | **58.23** | 58.36 | 48.03 | 53.44 | 52.51 | 55.38 | 52.68 |
| DAGMM | 53.48 | 55.16 | 55.81 | 50.43 | 51.76 | 52.45 | 49.41 | 47.63 | 50.86 | 48.79 | 48.86 | 51.22 |
| SOM-DAGMM | 51.39 | 52.92 | 54.56 | 50.88 | 52.31 | 51.79 | 52.22 | **50.89** | 49.53 | 50.22 | 49.97 | 53.82 |
| SOM | 55.21 | 57.69 | 56.39 | **54.80** | **56.87** | 57.42 | **58.44** | 48.46 | 56.64 | **57.67** | 58.32 | **65.43** |
| NeuTraLAD | 54.15 | 55.51 | 51.05 | - | 51.69 | 53.06 | 50.52 | - | 54.82 | 53.71 | 57.03 | - |
| LOE | 55.22 | 55.16 | 55.73 | - | 50.56 | 51.55 | 48.68 | - | **57.03** | 55.32 | **58.59** | - |
| LOF | 48.12 | 47.78 | 48.33 | - | 49.87 | 50.23 | 50.12 | - | 51.78 | 53.05 | 52.86 | - |
| OC-SVM | 42.94 | 43.35 | 43.05 | - | **51.75** | **52.34** | **51.87** | - | 51.68 | 50.42 | 51.12 | - |
| IF | 52.46 | 50.65 | 46.62 | - | 49.76 | 51.67 | 50.55 | - | 59.42 | 49.56 | 52.45 | - |
| RF | 55.32 | 58.53 | 52.37 | - | 50.22 | 50.76 | 50.17 | - | **98.65** | 92.35 | 66.87 | - |
| GB | **63.06** | **64.96** | **54.32** | - | 50.94 | 50.21 | 50.12 | - | 93.26 | **95.88** | **85.43** | - |

From Table 2, we observe that supervised methods outperform the unsupervised deep learning methods for our dataset and CI dataset but for VI dataset RSRAE performs better than shallow baseline methods. The supervised methods have the benefit of training with labels, which helps them distinguish between normal and anomalous samples. LOE uses contaminated training data, which is also followed in our work. This makes it more suitable to the setting and hence results in improved performance. SOM-DAGMM, which is an improvement of DAGMM because it introduces additional topological information to the estimation network, performs poorly than DAGMM in our case. Analogous to (Alvarez et al., 2022), where a simple deep autoencoder had the best performance

for tabular datasets, SOM also a simple model has comparable AUC scores to other models for all datasets and is also better in some cases as seen in Table2. RSRAE, which is state of the art for document and image datasets in unsupervised anomaly detection, performs better than other deep unsupervised methods for CI and VI datasets but has poor performance for the large dataset that is attributed to increased dimensionality due to categorical features. In LOE, which follows the strategy of using contaminated training data, GEL encoding is better than other encodings. If there is a wealth of training data available, SOM (competitive learning) approach is more suited for anomaly detection.

We use RSRAE, DAGMM, SOM-DAGMM, and SOM to compare the impact of the embedding layer with autoencoders. Due to the challenge of incorporating the embedding layer with NeuTral-AD and LOE, we did not use the models for this comparison. RSRAE performs better than other deep unsupervised methods for the CI dataset with other encodings but with embedding layer has poor results. Since we already know that training embeddings in unsupervised learning is challenging, in our work we observe the phenomenon empirically. Although, *Embedding* layer predicts more anomalies in SOM than other deep learning models and the improved performance in SOM-DAGMM using the Embedding layer can be attributed to the use of SOM in the initial stages. The difference between embeddings in SOM and other models is that they are not trainable in SOM and are a higher dimensional representation of the categorical values. This shows that the representation of categorical values in higher dimensions improves the performance of SOM but One Hot encoding encounters the curse of dimensionality. With the increase in the cardinality of the dataset, *One Hot* encoding should be avoided and the embedding layer is a viable alternative for encoding the categorical features. GEL encoding performs poorly with supervised baseline models in comparison to One Hot encoding.

The behavior of SOM with the embedding layer leads us to believe that it is possible to determine a critical dimension $d_c$ for encoding high cardinal features such that for an attribute with cardinality $N$, $d_c \ll N$ is the value such that for $k > d_c$ the unsupervised model performs poorly, where $k$ represents the dimension of the encoded attribute. (Zimek et al., 2012) mentions the challenges due to the curse of dimensionality in anomaly detection in his work that the subspace grows exponentially with dimensionality, therefore, making it inefficient to scan through the search space. Also, irrelevant features in the dataset can mask the relevant distances to distinguish between normal and anomalous samples. Due to the above reasons, we observe in Table 2 that the VC data performs better with Label and GEL encoding than One Hot Encoding for the deep unsupervised models. SOM is not affected by the increased dimensionality and outperforms all other approaches with the embedding layer in the case of Car Insurance and Vehicle Claims data. The preferred One Hot encoding is suited to small and medium datasets but the poor performance in the case of a large dataset makes it necessary to search for alternative encoding approaches. Our experiments with GEL and embedding layers depict their usefulness under different circumstances. GEL encoding works similarly to One Hot encoding although it consists of $k$ dimensions, where $k$ is the number of categorical attributes.

## 5   Conclusion

In this paper, we have contributed to the field of unsupervised anomaly detection by evaluating unsupervised anomaly detection approaches, SOM (Chen et al., 2021a) in competitive learning, RSRAE (Lai et al., 2020) in reconstruction-based methods, DAGMM (Zong et al., 2018) and SOM-DAGMM (Chen et al., 2021a) in density based estimation methods, NeuTral-AD (Qiu et al., 2021), LOE (Qiu et al., 2022) in contrastive learning for auditing datasets. Our evaluation suggests that for auditing datasets no single model or encoding is suitable for all datasets. Since low cardinal datasets performed better with One Hot encoding and high cardinal data performed poorly, we believe that instead of applying a uniform encoding method to all categorical features, we can use different encodings for features based on their cardinality. Ordinal features should be label encoded, low cardinal features should be One Hot encoded, and embeddings for high cardinal features will provide a better representation and remove the curse of dimensionality. Another issue related to unsupervised anomaly detection is finding an optimal threshold to separate normal and anomalous samples. The prior knowledge about the ratio of anomalies in the dataset does not always provide the best threshold. We report our findings on this later in A.6.

Our major contribution is creating a benchmark auditing dataset, i.e., Vehicle Claims, which is a large dataset containing 21 percent anomalous samples. It helps us to identify issues related to One Hot encoding, the preferred encoding method for categorical features in machine learning and deep learning models. Due to the high cardinality of features, we encounter the curse of dimensionality,

and the performance of Vehicle Claims data deteriorates across models. In contrast, Car Insurance and Vehicle Insurance data perform better with One Hot encoding in comparison to other encodings. The evaluation of our dataset shows that the features are relevant to the classification task as the supervised methods achieve more than 95% AUC score. Our aim was to introduce a public dataset that is useful for predicting anomalies in auditing data. We create the dataset based on the domain knowledge of fraudulent insurance claims and while evaluating the supervised and unsupervised methods it presents us with challenge of encoding categorical attributes. This is not a big factor in the case of small datasets but in the real world where we have migrated from data to big data, it will be incredibly helpful for ML practitioners.

The author implementation of NeuTral-AD[1] and LOE[2] are used while RSRAE, DAGMM and SOM-DAGMM are implemented to add the embedding layer to the models. The models were implemented in Pytorch. Minisom library is used for SOM[3]. For OC-SVM, Isolation Forest, LOF, Gradient boosting and Random Forest we use Scikit learn implementations. RSRAE[4] and DAGMM[5] are referenced from GitHub and modified to use embedding layer with the models.

## 6 Acknowledgements

MK acknowledges support by the Carl-Zeiss Foundation, the DFG awards KL 2698/2-1, KL 2698/5-1, KL 2698/6-1, and KL 2698/7-1, and the BMBF awards 03|B0770E and 01|S21010C.

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

# A  Appendix

## A.1  Datasets

### A.1.1  Vehicle Claims

In this section, we present the repair complexities assigned to automobile makers in our dataset in Table 4 and the number of hours and cost of repair for the associated repairs in Table 3.

Table 3: Time and cost of repairing vehicles based on the issue and issue_id. Small issues like Flat tyres require less amount of time to repair and cost lower than Engine Issues which require more repair time and amount. The values in the dataset reflect the time and amount approximately and not the original values.

| Issue | Issue Id | Time (Hours) | Repair Cost (Ratio of Price) |
|-------|----------|--------------|------------------------------|
| Brake Pads Worn | 1 | 2 | 0.0005 |
| Alternator Failing | 1 | 2 | 0.02 |
| Windscreen Crack | 1 | 1 | 0.0006 |
| Gear Box Issue | 1 | 2 | 0.01 |
| Flat Tyres | 1 | 1 | 0.0003 |
| Radiator Leaking | 1 | 2 | 0.02 |
| Excessive Emissions | 1 | 1 | 0.0009 |
| Steering Wheel Shaking | 1 | 1 | 0.001 |
| Tyre Alignment | 1 | 0.5 | 0.0001 |
| Starter Motor Issue | 1 | 3 | 0.01 |
| Sensor Malfunction | 1 | 3 | 0.05 |
| Electrical Issue | 5 | [2,0.5,1,2,3] | [0.001,0.002,0.005,0.003,0.001] |
| Warning Light | 8 | [1,0.5,2,5,3,2,1,9] | [0.001,0.005,0.003,0.005,0.004, 0.002,0.004.0.01] |
| Engine Issue | 4 | [8,16,12,10] | [0.2,0.15,0.1,0.05] |
| Transmission Issue | 3 | [1,2,8] | [0.003,0.007,0.009] |

Table 4: *repair_complexity* of respective makers in the Vehicle Claim dataset. The automotive makers which can be easily located and have more repair stations have lower *repair_complexity*.

| repair_complexity | Maker |
|---|---|
| 1 | Audi, BMW, Chevrolet, Dacia, Daewoo, Daimler, Fiat, Ford, GMC, Honda, Hyundai, Jeep, Kia, Lexus, Mazda, Mercedes-Benz, Mitsubishi, Nissan, Opel, Peugeot, Renault, SKODA, Santana, Smart, Suzuki, Toyota, Vauxhall, Volkswagen. |
| 2 | Brooke, Caterham, Citroen, DAX, DS, Abarth, Ginetta, Great Wall, Grinnall, Infiniti, Isuzu, Jensen, Koenigsegg, London Taxis International, MEV, MG, MINI, Morgan, Noble, Perodua, Pilgrim, Proton, Radical, Raw, Reva, SEAT, Saab, Sebring, Ssangyong, TVR, Tiger, Westfield, Zenos |
| 3 | Alfa Romeo, Aston Martin, Bentley, Cadillac, Chrysler, Daihatsu, Ferrari, Jaguar, KTM, Land Rover, Lincoln, Lotus, McLaren, Porsche, Rolls-Royce, Rover, Subaru, Volvo. |
| 4 | Corvette, Buggati, Dodge, Hummer, Lamborghini, Maserati, Maybach, Pagani, Tesla |

## A.1.2 Vehicle Insurance and Car Insurance

*Vehicle Insurance* dataset consists of 6 percent anomalous samples, which model real-world scenarios to a large extent. There are 26 categorical features and 6 numerical features. More categorical attributes than numerical ones is a desirable characteristic of an auditing dataset. Although categorical attributes are higher in number, but it is a medium-sized dataset with 15420 data points that leads to 151 unique categorical values in the dataset. This makes it unlikely that the data will suffer from the curse of dimensionality. *Car Insurance* dataset is a small-sized dataset available with labels and contains 25 percent anomalous samples. It consists of a similar number of numerical and categorical features. The number of unique categories in the complete dataset is 169. So, the dataset does not suffer from the curse of dimensionality either.

## A.2 Encodings

We briefly describe the GEL encoding and Embedding layers used in our work since they are uncommon categorical encoding approaches. One of the important characteristics of auditing datasets are the categorical features. *Which representation provides more information to the model*? If the values are ordinal, *Label* encoding is suitable for the feature. The most common approach is using *One Hot* encoding, but we observe that the model suffers from the curse of dimensionality for large datasets. Therefore, we investigate alternative approaches like *GEL* encoding and embedding layer (Guo and Berkhahn, 2016).

## A.2.1 GEL Encoding

GEL encoding is an embedding method that is applicable to both supervised and unsupervised learning approaches. Since it is based on the principle of One Hot encoding, we apply it to reduce the dimensionality of One Hot encoded categorical features. To obtain embedding features, GEL (Golinko and Zhu, 2019) uses class-based approach to separate data into different instance partitions. Then, GEL generates a row and feature representation of each instance partition. GEL transforms the data into a binary format such that it computes low-rank representation while training and use the same representation on new data. Our focus is on feature embedding for unsupervised tasks.

**Instance and Feature Marginal Distribution**

A matrix $\mathbf{W}$ is created by converting dataset $\mathbf{X}$ to binary representation. To obtain distribution of data in $\mathbf{W}$, two marginal distribution matrices $\mathbf{R} \in \mathrm{R}^{n \times 2}$ and $\mathbf{F} \in \mathbb{R}^{m_w \times 2}$, where n and $m_w$ denote number of rows and number of columns of $\mathbf{W}$ matrix, respectively.

Each row $\mathbf{R}_i \in \mathbf{R}$ corresponds to the marginal distribution of the instance $\mathbf{x}_i \in \mathbf{X}$ across its binary feature values. $\mathbf{R}_{i,0}$ denotes the percentage of binary features which have a value of 0, and $\mathbf{R}_{i,1}$

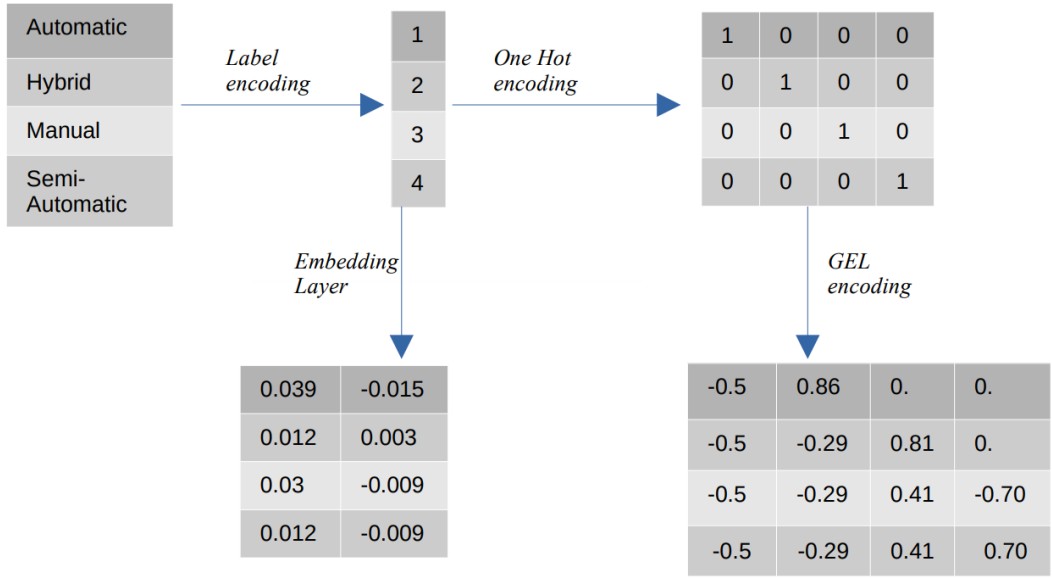

Figure 1: Representation of encodings and embeddings for an example containing four unique values.

denotes the percentage of binary features which have a value 1. Similarly, for the marginal distribution of each binary feature in $\mathbf{W}$ across all instances, $\mathbf{F} \in \mathbb{R}^{m_w \times 2}$ is used to calculate the percentage of binary values in the matrix $\mathbf{W}$.

$$\mathbf{R}_{i,0} = \frac{\sum_{i=1}^{j=m_w}(1 - w_{i,j})}{m_w} \tag{1}$$

$$\mathbf{R}_{i,1} = 1 - \mathbf{R}_{i,0} \tag{2}$$

$$\mathbf{F}_{i,0} = \frac{\sum_{i=1}^{j=n}(1 - w_{i,j})}{n} \tag{3}$$

$$\mathbf{F}_{i,1} = 1 - \mathbf{F}_{i,0} \tag{4}$$

**Instance-Feature Marginal Distribution Matrix Q**

To calculate instance and feature representation as one entity, an Instance-Feature marginal distribution matrix $\mathbf{Q}$ is defined.

$$\mathbf{Q} = \mathbf{F} \times \mathbf{R}^T \tag{5}$$

From the matrix, $\mathbf{Q}$, a low-rank representation $\mathbf{S}$ is created. Then, after singular value decomposition of $\mathbf{S} = \mathbf{U}\Lambda\mathcal{V}^T$, top $k$ eigenvectors are chosen from $\mathcal{V}$ corresponding to largest eigenvalues among $\Lambda$.

$$\mathbf{S} = \mathbf{Q}^T\mathbf{Q}\mathbf{W} \tag{6}$$

Here, we solve the issue of dimensionality encountered in one hot encoding by choosing k dimensions of the embedding feature representation matrix.

### A.2.2 Embeddings

In this section, we briefly describe the use of the embedding layer for categorical features. One of the first issues when applying machine learning or deep learning methods to text data is their representation and the representation should be semantically correct and the representations are meaningful. Word embeddings provide an efficient way to represent similar words with similar encodings. Embedding is a floating point value in form of a vector. Embeddings are trainable

---
**Algorithm 1** Generalized Feature Embedding Learning Algorithm. Golinko and Zhu (2019)
---
**Input**: $\mathbf{X}$: A dataset including categorical and numerical features.

**k**: Dimensionality of embedding feature space.

**Output**: $\mathcal{F} \in \mathbf{R}^{n \times k}$ : The learned embedding feature representation of instances in $\mathbf{X}$.

    1. Convert dataset $\mathbf{X}$ to a binary representation $\mathbf{W}$.

    2. Using Eqs. 2.1 - 2.4, calculate $\mathbb{F}$ and $\mathcal{R}$ matrix representations.

    3. Compute $\mathbf{Q}$ from Eq. 2.5.

    4. Compute $\mathbf{S}$ as in Eq. 2.6.

    5. $\mathbf{S} = \mathbf{U}\mathbf{\Lambda}\mathcal{V}^{T}$.

    6. $\mathcal{V} \in \mathbb{R}^{n \times n}$ = Eigenvectors of matrix $\mathbf{S}$.

    7. $\mathcal{V}^{k} \in \mathbb{R}^{n \times n}$ = First $k$ eigenvectors of $\mathcal{V}$ with largest eigenvalue magnitude.

    8. $\mathcal{F} = \mathbf{W}\mathcal{V}^{k}$, embedding feature representation of instances, where $\mathcal{F} \in \mathbb{R}^{n \times k}$.

    9. **return** $\mathcal{F}$.
---

parameters. It is common to see word embeddings that are 8-dimensional (for small datasets), and up to 1024-dimensions when working with large datasets. Embeddings convert each word present in the matrix to a vector with a properly defined size. The embedding layer is a lookup table where each word is converted to numbers, and the numbers are used to create the table. Thus, keys are represented by words, and the values are word vectors. Contrary to NLP where embedding is very helpful in reducing the dimensions, it increases the dimensions of our categorical features for better representation. Another advantage of embedding is that it connects words with contexts.

To generate dense word embeddings, the similarity is measured between multiple semantic attributes of the words. Each word has semantic attributes and a score is assigned to each attribute. Now we have a vector for each word with attribute scores. A similarity score such as cosine similarity might be used to check how close words lie to each other. Therefore, embeddings are a representation of the semantics of a word, efficiently encoding semantic information that might be relevant to the task at hand. In deep learning, instead of assigning the score to embeddings manually and calculating similarity scores, we let the neural network learn the embeddings depending on the dataset. Embeddings are parameters in our model and are updated during the training. After training, we have the learned embeddings that project similar words into the same space. We expect the learned representation of categorical features with an embedding layer to perform better than other encoding methods as they have shown better results in supervised tasks (Chen et al., 2016). We further explain the use of the embedding layer with the models in section A.4.

## A.3 Models

### A.3.1 SOM

Self Organizing Maps (SOM) are based on the principle of competitive learning. They were first introduced by (Kohonen, 1990) and have been recently used in the field of intrusion detection (Chen et al., 2021b). Unlike backpropagation, where the gradients of weights are used to optimize our model, in SOM the best matching unit (BMU) and the neighborhood radius is used to train the model. The neurons of the map are randomly initialized in the first step. Then, the distance between the input vector and the neuron determines the best matching unit i.e. the neuron with the smallest distance. The learning rate $\alpha = [0, 1]$ and neighbourhood radius $\sigma$ update the map. If the learning rate is 1, the BMU inherits the weights of the input vector as its weights. Similarly, $\sigma$ determines how many neurons in the neighborhood are updated, if the value is 2, the weights of neurons within 2 unit distance are updated. Both $\alpha$ and $\sigma$ values decay exponentially and decrease to 0 with an increasing number of iterations. In the end, the neurons for normal samples are closer than the anomalous ones. Anomaly scores are defined by the quantization error. The quantization error is the difference between the input sample and the weights of the best matching unit. Higher quantization error indicates a higher probability of the sample as an anomaly.

### A.3.2 RSRAE

The Robust Surface Recovery (RSR) layer (Lai et al., 2020) maps the latent representation from an encoder to a linear subspace which is robust to outliers. The objective function consists of two terms where the first is the reconstruction loss and the second term is the loss of RSR layer (Maunu and Lerman, 2019). Anomaly scores are defined by the reconstruction error. Higher reconstruction error indicates a higher probability of the sample as an anomaly.

### A.3.3 DAGMM & SOM-DAGMM

Deep Autoencoding Gaussian Mixture Model (Zong et al., 2018) is an energy-based model that defines anomalies as points that lie in a low-likelihood region. A compression network that consists of an autoencoder projects the input data to a lower dimension and then we use the latent representation and the reconstruction error and cosine similarity as input for the estimation network which is a neural network that estimates the parameters of gaussian mixture models. The objective function consists of three terms, reconstruction error, energy/likelihood of the point, and a regularizer to penalize the diagonal values in the covariance matrix. Self-organizing Map - Deep Autoencoding Gaussian Mixture Model (Chen et al., 2021a) is also an energy-based model. The working principle is the same as DAGMM. The only difference lies in the input to the estimation network where we also include the normalized coordinates of SOM for the input sample which provides the missing topological information in the case of DAGMM to the estimation network. The objective is also similar to DAGMM. Anomaly scores are defined by the estimated energy of the sample. Low energy indicates a higher probability of the sample as an anomaly.

### A.3.4 NeuTral-AD & LOE

NeuTral-AD (Qiu et al., 2021) consists of two components in the training pipeline. First, a set of learnable transformations and then an encoder. The encoder and transformations are jointly trained on the Deterministic Contrastive Loss (DCL) function. The goal of DCL is that the transformed sample should be similar to the original sample while the transformations should be dissimilar to other transformations of the sample. Latent Outlier Exposure (Qiu et al., 2022) considers the training of models with contaminated data, which is relevant to the real-world setting and our training strategy in this work. Loss of normal and anomalous data is optimized in the latent space. We use LOE with NeuTral-AD backbone for our experiments. The loss function is used for anomaly scoring in this case. Higher loss indicates an anomalous data sample.

### A.4 Embedding layer + Models

This section describes the use of embedding layers with existing unsupervised learning models. The reason for using the embedding layer is that with the introduction of categorical features, new challenges are encountered while encoding categorical features. We use *Label* encoding, *One hot* encoding, and *GEL* (Golinko and Zhu, 2019) encoding for categorical features. Embedding layer (Dahouda and Joe, 2021; Guo and Berkhahn, 2016) is also used in other contexts, such as risk classification (Shi and Shi, 2021) and time series forecasting (Karingula et al., 2021). (Guo and Berkhahn, 2016) demonstrated that entity embedding helped the neural network generalize in case of sparse data and unknown statistics. Embeddings are helpful in case of high cardinality of features. Using the embedding layer helps us compare the different encodings with embeddings and their impact on our model's ability to learn representations in an unsupervised setting.

Embeddings with neural networks perform better on supervised learning tasks. But in unsupervised tasks, the target is missing to train the embeddings. We can create embeddings by training models like FastText, and Word2Vec on entities, but there are two issues. *Firstly*, Embeddings from the above models are based on the assumption that words belong to a single feature. So, the generated embeddings are on the same plane. "Sunday" which is a day of the week, and "Yellow" a colour will lie on the same plane instead of separate planes. *Secondly*, In supervised learning, embeddings are trained with the help of a target, so they are mapped accordingly to the respective feature space for normal and anomalous samples. The embedding layers are trained with backpropagation and are used for categorical features.

The embedding can be efficiently used for training entities as shown by (Guo and Berkhahn, 2016). We try to learn embedding in an unsupervised way and compare it with other encoding strategies. The

embedding layer in the autoencoder (Ohi et al., 2020) is trained with the reconstruction loss. Training embeddings on an autoencoder does not require the target value because the network is trained with respect to the reconstruction loss computed between the input sample and decoder output.

### A.4.1 Embedding Layer with Autoencoders

In this paper, the embedding layer is used for encoding the categorical variables with RSRAE (Lai et al., 2020), DAGMM (Zong et al., 2018), and SOM-DAGMM (Chen et al., 2021a) for anomaly detection. We have already seen the use of entity embeddings for categorical variables (Guo and Berkhahn, 2016) and embedding layer (Ohi et al., 2020) with autoencoders for training purposes.

Let us assume that the input data $X \in \mathbb{R}^D$, has $k$ numerical features and $D - k$ categorical features. In general, we use *One Hot* encoding or *Label* encoding, whichever is more suitable to the dataset. We use the embedding layer from PyTorch for training our models. The size of the embeddings depends on the number of categories present in the feature.

$$d^e = \min(50, n/2) \tag{7}$$

Where $d_e$ is the embedding size of the feature and $n$ is the number of categories in the feature. We set the upper bound of embedding size to size 50 because for a feature with 500 unique values, we would have embeddings of size 250, which invokes the curse of dimensionality that we want to avoid while embedding. The input data is divided into $X^{1..k}$, numerical features, and $X^{(D-k)..D}$ categorical features. Then the categorical features are represented by embeddings $X^{emb}$ such that

$$x_j^{emb} = \sum_{j=1, i=(D-k)}^{j=1n, i=D} E(d_j^e, x_{j,i}) \tag{8}$$

where $E(\cdot)$ is the embedding layer, $d_j^e$ is the embedding size for $i^{th}$ feature, and $n$ is the number of data points. The embeddings $X^{emb}$ and numerical features are combined, $X = [X^{emb}, X^{1..k}]$ and then passed as input to the model. The embedding layer representations are used as input to the autoencoder and trained on the reconstruction loss or the model's objective function in the above models.

### A.4.2 Embedding Layer with Self-Organizing Maps

Similarly, we used embedding layer representations for training Self Organizing maps. There is no previous literature regarding the use of embeddings with SOM. The embedding layer is not trainable in this case and is a high-dimensional representation of the features.

### A.5 Experimental Setting

We test SOM (Chen et al., 2021a), RSRAE (Lai et al., 2020), DAGMM (Zong et al., 2018), and SOM-DAGMM (Chen et al., 2021a) on *Vehicle Claims*, *Vehicle Insurance* and *Car Insurance* with *One Hot*, *Label*, *GEL* encodings, and *Embedding* layer.

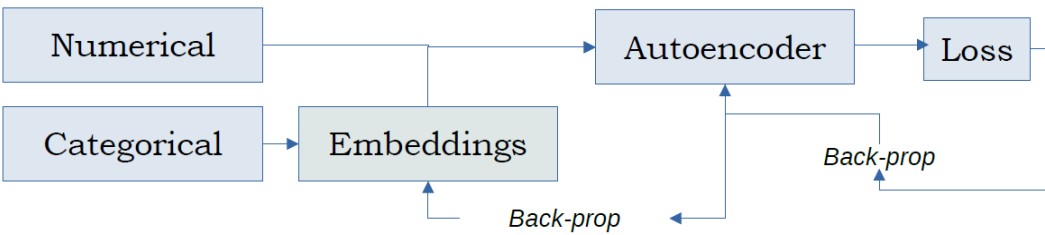

Figure 2: Trainable embedding layer with autoencoder (Ohi et al., 2020). The embeddings are updated via backpropagation.

### A.5.1 RSRAE

For all the datasets, the encoder consists of three fully connected linear layers in case of encodings and an extra linear layer in case of embeddings. The encoder reduces the data to $D$ dimensions where $D = 4$ are used in the setting. The output of the encoder is flattened, and the RSR layer transforms it into a d-dimensional vector. We use $d = 2$ in the experiments for all datasets. The decoder consists of a dense layer that transforms the output of the RSR layer into a vector of the same shape as the output of the encoder and then three fully connected linear layers that reconstruct the output of the RSR layer into the shape of the input data. LeakyRELU is used as the activation function. $\lambda_1$ and $\lambda_2$ are the meta parameters in RSRAE. In the paper (Lai et al., 2020) and this work, $\lambda_1 = 0.1$ and $\lambda_2 = 0.1$ are used. We train using batch size = 128, 256 with similar results. Adam is used for optimization, and the learning rate is 0.001.

### A.5.2 DAGMM & SOM-DAGMM

The encoder consists of two fully connected linear layers for the compression network that map the input data to $D$ dimensions where $D = 4$ is used. The estimation network consists of a simple neural network consisting of two fully connected linear layers with $tanh()$ activation for the first layer and $softmax$ activation for the output layer. The input dimension to the estimation network is $D + 2$, in our case, because we use reconstruction loss and cosine similarity as additional features similar to usage in the paper. The output dimension is the number of gaussian mixtures $G = 2$. In the case of SOM-DAGMM, all the parameters are the same except the input to the estimation network, which is $D + 4$, as we add the coordinates of the SOM winner neuron. $\lambda_1$ and $\lambda_2$ are the meta parameters in DAGMM. In the paper, $\lambda_1 = 0.1$ and $\lambda_2 = 0.005$ were used. In our work, we use $\lambda_1 = 1$ and $\lambda_2 = 0.05$ because of the degeneracy issue mentioned in DAGMM, due to which we had to deal with infinity values. Training DAGMM was challenging because, for $batchsize = 128, 256$, the loss of the model would become *Nan* or *Infinity*. The batch size used for the *Vehicle Claim* dataset is $85, 159$, which is a factor of the total number of rows in the training dataset. For other datasets we use batch size = 128, 256. Adam is used for optimization, and the learning rate is 0.001.

### A.5.3 SOM

For training of SOM, we use different sizes of maps ranging from $5 \times 5$ to $25 \times 25$ with an increment of size 5. The learning rate $\alpha$ for SOM in the computation is 0.5, and the $\sigma$ value for the neighborhood is also 0.5. The batch size for all datasets is 128.

### A.5.4 NeuTral-AD & LOE

We use 9 transformations, and the size of the hidden layer is 16 for our datasets. The batch size is 128, and Adam optimizer is used. Other parameters are the same as the official implementation of the paper.

### A.6 Constrained Threshold Evaluation

Here we report the evaluations based on F1 scores. The reason for using two different metrics for evaluation is that AUC is unaffected by an imbalance in the distribution of classes, and it gives equal weights to both normal and anomalous samples in the prediction. AUC is calculated by varying the threshold $\alpha$ from $-\inf to +\inf$. Therefore, we use AUC because it gives us a better idea of the model's performance regardless of the threshold. We report F1 scores to choose the optimum threshold for evaluation. *What is the optimal threshold*? The threshold at which the normal and anomalous samples can be distinguished easily. A common approach is to select $\alpha$ based on the ratio of normal samples in the dataset. If $\tau$ is the ratio of anomalies present in the dataset, then $\alpha = 1 - \tau$ is used for selecting the threshold to distinguish between normal and anomalous samples. We use F1-Score to evaluate our models between $\tau = 0.05$ and $\tau = 0.30$, as it is observed in Table 1 that is the range of the ratio of anomalies in the datasets used for evaluation.

We observe from Figure 3 that the F1 score is decreasing as the anomaly ratio threshold increases for the VI dataset. This is due to weighted F1-score metrics used for evaluation where the metrics are calculated for each class, and the average is weighted according to the number of true samples for each class. It contains 6 percent anomalous samples; therefore the negative class is associated with

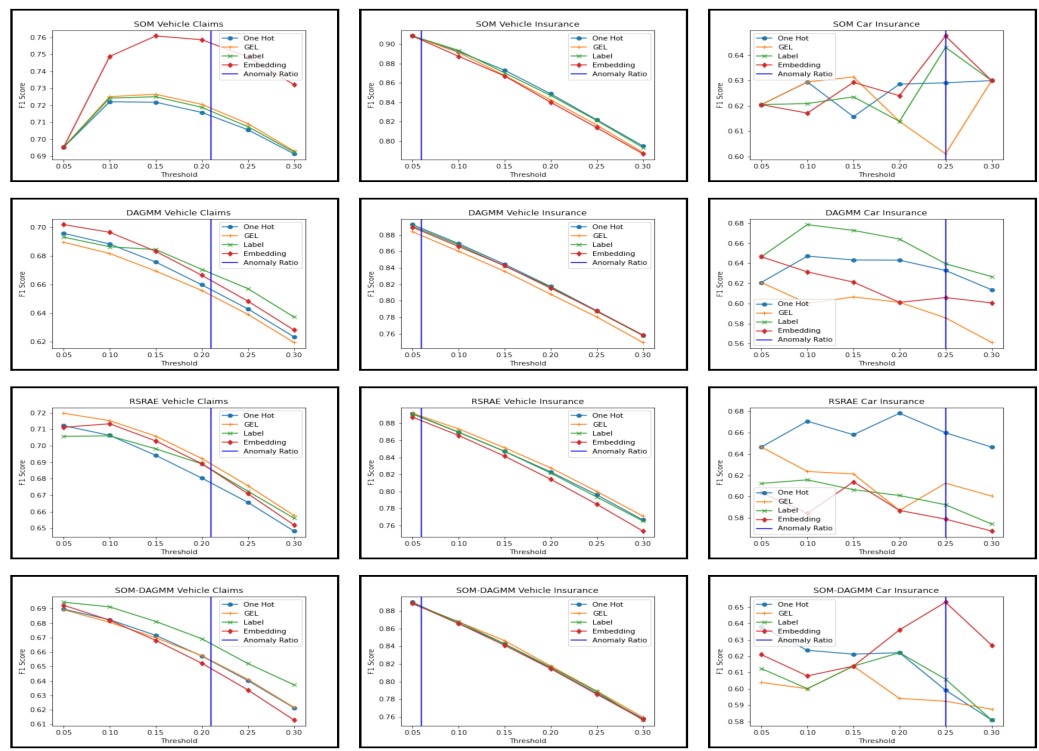

Figure 3: The *vertical* line represents the ratio of anomalies present in the dataset. If we select the threshold at the ratio of anomalies present, the best scores should be reported at the vertical line in the above plots, but it occurs only for a few instances of Car Insurance data.

a larger weight than the positive class in comparison to *Vehicle Claims* and *Car Insurance* datasets where they have 21 and 25 percent anomalies. As the threshold is decreased, the number of true negatives increases because we consider less percentage of data to be anomalous, and the negative class is correctly classified by the model. Alternatively, when we increase the threshold, the number of true positives increases. Assuming the model should perform best for the threshold equal to the anomalies percentage, it should perform best at 0.25 anomaly ratio for the CI data and at 0.21 for the VC data. But in an unsupervised scenario, without prior information about the percentage of anomalous samples, it becomes difficult to select a threshold to distinguish between normal data and anomalies. In the context of insurance claims, we want to reduce the number of false negatives as well as false positives. Fraudulent claims cannot be missed and Normal claims marked as anomalies increase the operational cost due to verification by an expert. Therefore, we need to find an optimal threshold for unsupervised learning scenarios to reduce the type 1 and type 2 errors. Currently, the threshold is decided by domain experts based on historical information, but different datasets are contaminated with dissimilar proportions of anomalies which makes it challenging to select the threshold for evaluation.

