# OpenReview forum: "Unsupervised Anomaly Detection for Auditing Data and Impact of Categorical Encodings."
_NeurIPS.cc/2022/Workshop/SyntheticData4ML — Neurips 2022 SyntheticData4ML_

### Official Review · Reviewer_oU5R · 2022-10-18
**Interesting dataset and promising results**

**Rating:** 6
**Confidence:** 3

**Review:**

This paper introduces Vehicle Claims datasets which belong to the category of Auditing data that includes Journals, Insurance claims, and Intrusion data for information systems. It consists of fraudulent insurance claims for automotive repairs. This paper tackles the problem of missing benchmark datasets for anomaly detection as the datasets are mostly confidential, and the public tabular datasets do not contain relevant and sufficient categorical attributes. Therefore, a large sized dataset is created for this purpose and referred to as Vehicle Claims (VC) dataset. The dataset is evaluated on shallow and deep learning methods.

This paper presents interesting dataset and promising results.

---

### Official Review · Reviewer_LeyC · 2022-10-18
**Good evaluation of different anomaly detection techniques and encoding; synthetic dataset generation methodology is not generalizable**

**Rating:** 6
**Confidence:** 3

**Review:**

The paper introduces a synthetic benchmark dataset which is helpful in the context of auditing models as datasets are not widely available. Additionally, the paper evaluates a bunch of state-of-the-art techniques on different benchmark datasets, along with encoding techniques. The authors make good data-driven remarks on when different techniques becomes appropriate, which will help out future researchers. The synthetic dataset, however, was created using a very particular method that involves another dataset, which is not generalizable.

---

### Meta-Review · Area_Chair_fDZV · 2022-10-20

**Recommendation:** Accept